# Computational network biology analysis revealed COVID-19 severity markers: Molecular interplay between HLA-II with CIITA

Heewon Park[1,2,3]*, Satoru Miyano[2,3]

**1** School of Mathematics, Statistics and Data Science, Sungshin Women's University, Seoul, Republic of Korea, **2** M&D Data Science Center, Institute of Science Tokyo, Bunkyo-ku, Tokyo, Japan, **3** Human Genome Center, The Institute of Medical Science, The University of Tokyo, Minato-ku, Tokyo, Japan

* hwpark@sungshin.ac.kr

**Data availability statement:** The datasets are available at the National Bioscience Database Center (NBDC) Human Database (accession code: hum0343.v2).

**Funding:** This research was also supported by AMED under Grant Number 23tk0124003h0001, 24tk0124003h0002, JSPS KAKENHI Grant Number 24H00009/22H03692 and JST Grant Number JPMJCR20H2 to S.M. The funders had

## Abstract

COVID-19, severe acute respiratory syndrome coronavirus 2, rapidly spread worldwide. Severe and critical patients are expected to rapidly deteriorate. Although several studies have attempted to uncover the mechanisms underlying COVID-19 severity, most have focused on the perturbations of single genes. However, the complex mechanism of COVID-19 involves numerous perturbed genes in a molecular network rather than a single abnormal gene. Thus, we aimed to identify COVID-19 severity-specific markers in the Japanese population using gene network analysis. In order to reveal the severity-specific molecular interplays, we developed a novel computational network biology strategy that measures dissimilarity between networks based on the comprehensive information of gene network (i.e., expression levels of genes and network structure) by using Kullback–Leibler divergence. Monte Carlo simulations demonstrated the effectiveness of our strategy for differential gene network analysis. We applied this method to publicly available whole blood RNA-seq data from the Japan coronavirus disease 2019 Task Force and identified differentially regulated molecular interplays between 368 severe and 105 non-severe samples. Our analysis suggests the gene network between HLA class II, CIITA, and CD74 as a COVID-19 severity specific molecular marker. Although the association between HLA class II and COVID-19 has been demonstrated, our data analysis revealed that the molecular interplay of HLA class II with its target and/or regulator is a crucial marker for COVID-19 severity. Our findings from computational network biology analysis suggest that suppression and activation of the molecular interplay between HLA class II, CIITA, and CD74 provide crucial clues to uncover the mechanisms of COVID-19 severity.

## Introduction

The nature and severity of coronavirus disease 2019 (COVID-19) differs significantly between individuals and populations [1]. While the exact determinants of severe disease are not

no role in study design, data collection and analysis, decision to publish, or preparation of the manuscript.

**Competing interests:** The authors have declared that no competing interests exist.

well-defined, current evidences suggest that host factors play a more significant role in driving pathogenesis than viral genetic mutations [2].

To uncover the complex mechanism underlying COVID-19 severity, several studies, especially focused on gene expression levels analysis, have been conducted to identify severity-associated markers. Ren et al. [3] identified key genes that could be used to distinguish between different phases of COVID-19, which is, the healthy, moderate, severe, and convalescent phases, i.e., they demonstrated that the gene markers, such as PFN1, RPS26, and FTH1, played key roles in severe acute respiratory syndrome coronavirus 2 (SARS-CoV-2) infection. Li et al. [4] uncovered markers for differentiating COVID-19 from common inflammatory responses, non-COVID-19 severe respiratory diseases, and healthy populations based on single-cell profiling of gene expression in six immune cell types. In their study, IFI44L in B cells, S100A8 in monocytes, and NCR2 in natural killer cells were identified as crucial markers that are involved in the innate immune response of COVID-19, while it was also demonstrated that ZFP36L2 in CD4+ T cells can regulate the inflammatory process of COVID-19 [4]. Peterson et al. [5] analyzed peripheral blood gene expression patterns and identified more than 6,000 differentially expressed genes between severe and non-severe illness, where most (85%) of the differentially expressed genes were under expressed, particularly with a significant impact on lymphocytes and changes in their function. Furthermore, many studies have been conducted to reveal the inflammatory mechanisms involved in COVID-19 [6,7], because the inflammation management is a promising strategy for addressing COVID-19 [8]. The association of genetic risk of severe COVID-19 with low inflammatory marker levels was also revealed by the genome-wide association study (GWAS) of SARS-CoV-2-negative cohort [9]. Furthermore, various studies on computational strategy for understanding infections have been also conducted [10–12].

Although previous studies performed to uncover mechanism underlying COVID-19, most have focused on gene expression levels and only on the perturbations of single genes, i.e., differentially expression gene analysis that is one of the widely used techniques for RNA-sequencing (RNA-seq) data analysis for identifying differentially expressed genes across two or more phenotypes [13], even though the complex mechanism of COVID-19 involves several genes that are connected in a molecular network rather than a single abnormal gene [14].

In an effort to uncover the mechanism underlying COVID-19 severity, we aimed to focus on identifying the COVID-19 severity-specific molecular interplay. We developed a novel computational strategy, called a Differentially regulated Gene Network detector (DGNdetector), to identify the differentially regulated gene network between severe and non-severe samples of COVID-19. We performed differential gene network analysis based on comprehensive information on gene networks, which is not only the expression levels of genes, but also the gene network structure by using the Kullback–Leibler divergence that is the dissimilarity measure of probability density functions [15]. The incorporation of the comprehensive information of gene network enables us to effectively identify differentially regulated gene networks.

We demonstrated the performance of our strategy for differential gene network analysis using Monte Carlo simulations in various scenarios. The DGNdetector was evaluated by comparison with existing methods SAM-GS [16] and GSCA [17] for different network identification. Our strategy showed outstanding performance in various scenarios of network structures, mean and variance of expression levels of genes.

We applied our method to whole-blood RNA-seq data obtained from the Japan COVID-19 Task Force, where the dataset is publicly available in https://www.nature.com/articles/s41467-022-32276-2#data-availability [18]. For 368 severe and 105 non-severe samples, we estimated COVID-19 severe and non-severe gene networks, respectively. We focused on

the identified more than six hundred genes by the Japan COVID-19 Task Force based on expression quantitative trait loci (eQTLs), splice QTLs (sQTLs) and differential gene expression analysis [18]. The DRGdetector succeeded in revealing 15 differentially regulated gene networks of the markers. Our results showed that the severe samples have relatively complex gene networks composed of larger number of genes and edges compared with those of the non-severe samples. The identified one of the largest networks consists of the interplays between Human Leukocyte Antigen (HLA) class II gene family, i.e., HLA class II dominated the identified differentially regulated gene network in COVID-19 severe samples, while the interplays became weaker and/or disappeared in that of non-severe samples. That is, the interplays between HLA class II gene family can be considered as severe specific characteristic. The interplays between the XCL family was also demonstrated as a COVID-19 severity-specific molecular interplay. In particular, the identified network comprising HLA class II, CIITA, and CD74 likely played a role in COVID-19 severity, i.e., it was demonstrated that the antiviral activities of CIITA and CD74 protect against coronaviruses [19]. We also revealed that the identified severe specific gene network is enriched in "hsa04612: Antigen processing and presentation". Genes in the identified subnetwork were verified in the literature. Although the association between HLA class II and COVID-19 has been uncovered in many previous studies [20–22], our study revealed that the molecular interplay between HLA class II and CIITA/CD74 is a key marker underlying the severity of COVID-19. Our results suggest that the suppression or activation (or both) of the interplay between HLA class II, CIITA, and CD74 may provide crucial clinical insights for understanding the mechanism of COVID-19 severity.

The novelties of our study are trying to uncover COVID-19 severe specific biomarkers based on the molecular interplays not abnormalities of each single gene and developing a novel computational strategy for identifying differentially regulated gene networks between COVID-19 severe and non-severe samples. Our strategy revealed COVID-19 severe specific molecular interplays between HLA class II and CIITA/CD74, those cannot be revealed by the single gene-based existing studies. The identified severe-specific molecular interplays may provide vital clues to uncover severe COVID-19 mechanism, because the complex mechanism of severe COVID-19 involved with numerous perturbed genes in the molecular networks rather than a single abnormal gene.

## Computational network biology strategy for revealing COVID-19 severity specific molecular interplay

Suppose $X = (x_1, \ldots, x_n)^T \in \mathbb{R}^{n \times p}$ and $Y = (y_1, \ldots, y_n)^T \in \mathbb{R}^{n \times p}$ are $n \times p$ data matrices that describe the expression levels of $p$ genes for $n$ samples in phenotype A (e.g., severe samples) and B (e.g., non-severe samples), respectively. Like previous studies [23,24], we assumed that the gene expression levels of each sample, $x_i = (x_{i1}, \ldots, x_{ip})^T$ and $y_i = (y_{i1}, \ldots, y_{ip})^T$, are independent and follow a Gaussian distribution $N(\mu_A, \Sigma_A)$ and $N(\mu_B, \Sigma_B)$, respectively, where $\mu_A$ ($\mu_B$) is the mean vector and $\Sigma_A$ ($\Sigma_B$) is a $p \times p$ covariance matrix of the expression levels of genes $X$($Y$). The precision matrices $\Omega_A = \Sigma_A^{-1}$ and $\Omega_B = \Sigma_B^{-1}$, that describe the dependence network structures between genes, are positive definite and symmetric matrices that are used to describe a weighted graph $G = (V, E, W)$, where $V$ is the set of vertices corresponding to $p$ genes, $E \in V \times V$ is the set of edges, where $(i, j) \in E$ indicates a link between vertices $i$ and $j$ (i.e., $i^{th}$ and $j^{th}$ genes). $W = (w_{ij})$ is the edge weight between vertices $i$ and $j$. The gene network can be represented by a weighted directed graph $G$ [25].

### Previous studies for differential gene set and network analysis

Significance analysis of gene expression profile for gene sets (SAM-GS)

Dinu et al. [16] proposed a method called the significance analysis of microarray for gene sets (SAM-GS) to identify significantly expressed gene sets in specific phenotypes. The SAM-GS measures the difference in expression levels between phenotypes based on the following statistics:

$$D_{\text{SAM-GS}} = \sum_{j \in V} \frac{(\bar{x}_j - \bar{y}_j)^2}{s_j + s_0}, \tag{1}$$

where $\bar{x}_j$ and $\bar{y}_j$ are the averages of the expression levels of the $j^{th}$ gene in phenotypes A and B, respectively; $s_0$ is a tuning parameter; and $s_j$ is the following gene-specific scatter:

$$s_j = \sqrt{a\{\sum_{i=1}^{n_A}(x_{ij} - \bar{x}_j)^2 + \sum_{k=1}^{n_B}(y_{kj} - \bar{y}_j)^2\}}, \tag{2}$$

where $n_A$ and $n_B$ are the numbers of samples in phenotypes A and B, respectively, and $a = (1/n_A + 1/n_B)/(n_A + n_B - 2)$.

Gene set co-expression analysis (GSCA)

A gene set co-expression analysis (GSCA) methodology was also developed to identify differentially co-expressed genes [17]. GSCA computes pairwise correlations of all $\binom{|V|}{2}$ gene pairs in network $G$, where $|V|$ is the number of genes. The statistic of the GSCA measures the dispersion of correlations between phenotypes A and B, as follows:

$$D_{\text{GSCA}} = \sqrt{\frac{1}{|V|(|V| - 1)/2} \sum_{k=2}^{|V|} \sum_{j=1}^{k-1} (c_{kj}^A - c_{kj}^B)^2}, \tag{3}$$

where $c_{kj}^A$ and $c_{kj}^B$ are the correlations between the $k^{th}$ and $j^{th}$ genes for phenotypes $A$ and $B$, respectively.

Although existing methods can identify responsive gene sets and networks that characterize a specific phenotype, they are not sufficient to effectively identify differentially regulated gene networks because the methods are based only on the expression levels of genes without considering the gene network structure.

### Differentially regulated gene network detector based on Kullback–Leibler divergence

We developed the DGNdetector to reveal responsive gene networks by incorporating simultaneously information of gene expression levels and the network structure. This is the novelty of our method. In order to incorporate comprehensive information about gene network, i.e., expression levels and network structure, we proposed the use of the Kullback–Leibler divergence for measuring dissimilarity of gene networks based on mean vectors of expression levels of genes and precision matrices. The Kullback–Leibler divergence measures the closeness between the probability distribution functions. For continuous models, the Kullback–Leibler divergence of two probability distributions with density functions $g(x)$ and $f(x)$ is defined as

follows [15]:

$$KL(g;f) = \int_{-\infty}^{\infty} \log\{\frac{g(x)}{f(x)}\}g(x)dx, \tag{4}$$

and has the following properties:

$$KL(g;f) \geq 0 \quad \text{and} \quad KL(g;f) = 0 \Leftrightarrow g(x) = f(x). \tag{5}$$

The gene expression levels $\boldsymbol{x} \sim N(\mu_A, \Sigma_A)$ and $\boldsymbol{y} \sim N(\mu_B, \Sigma_B)$ have the following probability density functions,

$$f(\boldsymbol{x}|\mu_A, \Sigma_A) = \frac{1}{(\sqrt{2\pi})^p}\exp\left[-\frac{1}{2}(\boldsymbol{x} - \mu_A)^T\Sigma_A(\boldsymbol{x} - \mu_A)\right], \tag{6}$$

$$g(\boldsymbol{y}|\mu_B, \Sigma_B) = \frac{1}{(\sqrt{2\pi})^p}\exp\left[-\frac{1}{2}(\boldsymbol{y} - \mu_B)^T\Sigma_B(\boldsymbol{y} - \mu_B)\right],$$

where $\mu_A$ and $\mu_B$ are $p$ dimensional mean vectors of expression levels of genes, and $\Sigma_A = \Omega_A^{-1}$ and $\Sigma_B = \Omega_B^{-1}$ are $p \times p$ covariance matrices (i.e., inverse precision matrices) that played to describe the gene network.

In order to measure dissimilarity of gene networks based on not only expression levels of genes (i.e., $\mu_A$ and $\mu_B$) but also network structure (i.e., $\Sigma_A = \Omega_A^{-1}$ and $\Sigma_B = \Omega_B^{-1}$), we measure the closeness of the probability density functions: $f(\boldsymbol{x}|\mu_A, \Sigma_A)$ and $g(\boldsymbol{y}|\mu_B, \Sigma_B)$ based on the Kullback–Leibler divergence as follows:

$$KL = \int_{-\infty}^{\infty} \log\left\{\frac{f(\boldsymbol{x}|\mu_A, \Sigma_A)}{g(\boldsymbol{y}|\mu_B, \Sigma_B)}\right\}f(\boldsymbol{x}|\mu_A, \Sigma_A)dx, \tag{7}$$

$$= \frac{1}{2}\left(\log\frac{|\Sigma_A|}{|\Sigma_B|} - p + \text{tr}(\Sigma_B^{-1}\Sigma_A) + (\mu_A - \mu_B)^T\Sigma_B^{-1}(\mu_A - \mu_B)\right).$$

As shown in the properties in (5), the Kullback–Leibler divergence is always positive and is zero if the two distributions are identical, i.e., $f(\boldsymbol{x}|\mu_A, \Sigma_A) = g(\boldsymbol{y}|\mu_B, \Sigma_B)$, and larger otherwise. This implies that the gene network corresponding to the large value of the *KL* can be considered as a differentially regulated gene network between phenotypes A and B, because the *KL* value indicates that expression levels of the genes (i.e., $\mu_A$ and $\mu_B$) in the gene network and/or network structure (i.e., $\Sigma_A$ and $\Sigma_B$) have large differences between phenotypes A and B.

To assess the significance of gene network dissimilarity, we considered the permutation framework and computed the permutation p-value of the Kullback–Leibler divergence. First, we generated permutation samples for phenotypes A and B (i.e., $X^{pm}$ and $Y^{pm}$, $pm = 1, \ldots, T$), and then estimated gene networks based on the permutation samples. In other words, we estimated the permutation precision matrices $\Omega_A^{pm} = \Sigma_A^{-1}$ and $\Omega_B^{pm} = \Sigma_B^{-1}$ for $pm = 1, \ldots, T$. We then computed the Kullback–Leibler divergence to measure the dissimilarity of the permutation networks as follows:

$$KL^{pm} = \frac{1}{2}\left(\log\frac{|\Sigma_A^{pm}|}{|\Sigma_B^{pm}|} - p + \text{tr}((\Sigma_B^{pm})^{-1}\Sigma_A^{pm}) + (\mu_A^{pm} - \mu_B^{pm})^T(\Sigma_B^{pm})^{-1}(\mu_A^{pm} - \mu_B^{pm})\right). \tag{8}$$

The permutation p-value was computed as follows [26]:

$$\text{p.value} = \frac{\sum_{pm=1}^{T} \boldsymbol{I}(KL \leq KL^{pm})}{T},$$ (9)

where $\boldsymbol{I}(\cdot)$ is an indicator function and $T$ is the number of permutations.

The Kullback–Leibler divergence measures the dissimilarity between probability density functions defined by the mean of the expression levels and network structure (i.e., inverse precision matrix). This implies that our strategy incorporates comprehensive information on gene networks, which is, the mean of expression levels and gene network structure, and it leads to biologically reliable result for differential gene network analysis.

Fig 1 shows the overall framework of our strategy for differentially regulated gene networks.

## Motel Carlo simulations for evaluating the DGNdetector

Monte Carlo simulations were performed to investigate the performance of the proposed strategy. We assumed two phenotypes A/B, and 10 subnetworks consisting of five common subnetworks for two phenotypes and five phenotype A/B-specific subnetworks. Each subnetwork consisted of 10 genes, and their regulatory network structures (i.e., precision matrices) were randomly generated using the *huge.generator* function in the R package Huge.

### Simulation study 1

In scenario 1, we first generated five precision matrices $\Omega_c^{nw}$ for $nw = 1, \dots, 5$ from "scale-free" graph structure for common subnetworks, whereas we generated the precision matrices of phenotype A and B -specific networks $\Omega_A^{nw}$ and $\Omega_B^{nw}$ for $nw = 1, \dots, 5$ with "scale-free" and "band" graph structures, respectively (see Fig 2). We then generated the expression levels of the five common subnetworks from $N(\boldsymbol{0}_{10}, (\Omega_c^{nw})^{-1})$ for $n = 100$ samples. The expression levels of phenotype A and B-specific networks (i.e., $\boldsymbol{X}$ and $\boldsymbol{Y}$) were generated from $N(\mu_A, (\Omega_A^{nw})^{-1})$ and $N(\mu_B, (\Omega_B^{nw})^{-1})$ for $n_A = n_B = 50$, respectively. For scenarios 2, 3, and 4, we generated expression levels similar to scenario 1, except for the precision matrices $\Omega_c^{nw}$ and $\Omega_A^{nw}$ for $nw = 1, \dots, 5$, which were generated with "random", "hub" and "cluster" graph structures, respectively. Fig 2 shows the graph structures for scenarios 1-4, where networks in yellow, black and grey boxes are common, pheynotype A and B -specific subnetwork, respectively. gene networks were estimated using a graphical lasso [27] based on the generated expression levels.

To identify differentially regulated gene networks, we performed a permutation test based on the number of permutations $T = 500$ and $p.value < 0.01$. Our method was evaluated using accuracy, recall, precision (PREC), true negative rate (TNR), and F-measure by comparing it with the existing methods SAM-GS and GSCA. We also considered anther approach for the differentially regulated gene network identification that measures the dissimilarity of two graphs based on the eigen values of the adjacency matrices as follows [28],

$$d(G^A, G^B) = \sqrt{\sum_{r=1}^{q} (\lambda(W(G^A))_r - \lambda(W(G^B))_r)^2},$$ (10)

where $q$ is the number of eigenvalues for the rank approximation of $W(G^A)$ and $W(G^B)$, and $\lambda(W(G^Y))_r$ and $\lambda(W(G^N))_r$ are $r^{th}$ eigenvalues of the weighted adjacency matrices of $W(G^A)$

**Expression levels of genes**

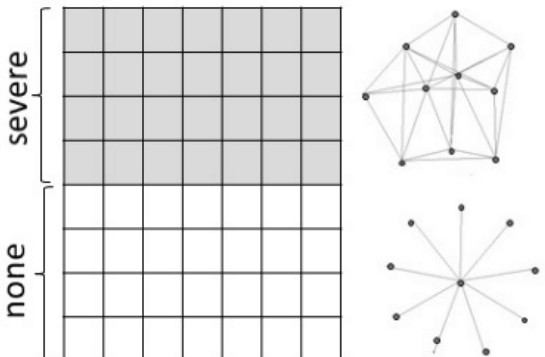

**Permutated expression levels of genes**

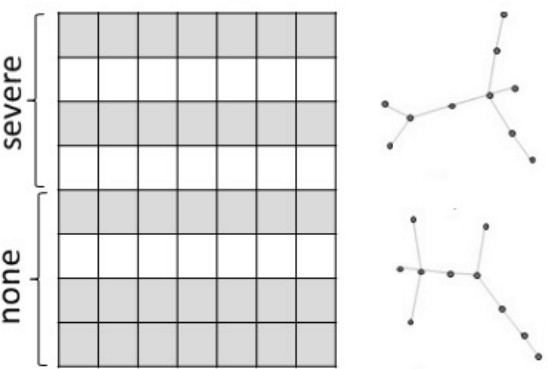

**Kullback–Leibler divergence**

$$KL = \int \log \frac{f(\boldsymbol{x}|\boldsymbol{\mu}_A, \boldsymbol{\Sigma}_A)}{g(\boldsymbol{y}|\boldsymbol{\mu}_B, \boldsymbol{\Sigma}_B)} f(\boldsymbol{x}|\boldsymbol{\mu}_A, \boldsymbol{\Sigma}_A) dx$$

**Permutated Kullback–Leibler divergence**

$$KL^{pm} = \int \log \frac{f\left(\boldsymbol{x}^{pm}|\boldsymbol{\mu}_A^{pm}, \boldsymbol{\Sigma}_A^{pm}\right)}{g\left(\boldsymbol{y}^{pm}|\boldsymbol{\mu}_B^{pm}, \boldsymbol{\Sigma}_B^{pm}\right)} f\left(\boldsymbol{x}^{pm}|\boldsymbol{\mu}_A^{pm}, \boldsymbol{\Sigma}_A^{pm}\right) dx$$

**Permutation p.value**

$$p.value = \frac{\sum_{pm=1}^{T} I(KL < KL^{pm})}{T}$$

**Fig 1. Overall framework of the differential gene network analysis.** Gene networks were estimated by severe and non-severe samples, and then Kullback–Leibler divergence was computed. The permutation samples for two groups were generated and then permutation networks were estimated. For the permutation gene networks, permutation Kullback–Leibler divergence was also computed. We detect differentially regulated gene networks based on the permutation p.value of Kullback–Leibler divergence.

and $W(G^B)$ for phenotypes A and B, respectively. We considered various situations for the mean and variance (i.e., the diagonal entries $\sigma^2$ of $\Sigma_A$ and $\Sigma_B$) of the expression levels to consider the realistic situations: there are not a difference of mean of expression levels between severe and non-severe samples and large variance of expression levels.

**Situation 1** The difference in the mean of the expression levels between the two phenotypes is not large (i.e., $\mu_A = 0, \mu_B = 0.3$) and $\sigma = 1$.

**Situation 2** There is considerable difference in the mean of expression levels between the two phenotypes: $\mu_A = 0, \mu_B = 1, \sigma = 1$.

**Situation 3** There is a considerable difference in the mean of expression levels between the two phenotypes and a large variance in the expression levels: $\mu_A = 0, \mu_B = 1, \sigma = 3$.

Table 1 shows the differentially regulated gene network identification results.

As shown in Table 1, the methods show similar results when the two phenotypes have different expression levels with $\sigma = 1$ (i.e., situation 2: $\mu_A = 0, \mu_B = 1, \sigma = 1$; see the center of Table 1), whereas the GSCA shows poor results for a large variance of expression levels (i.e.,

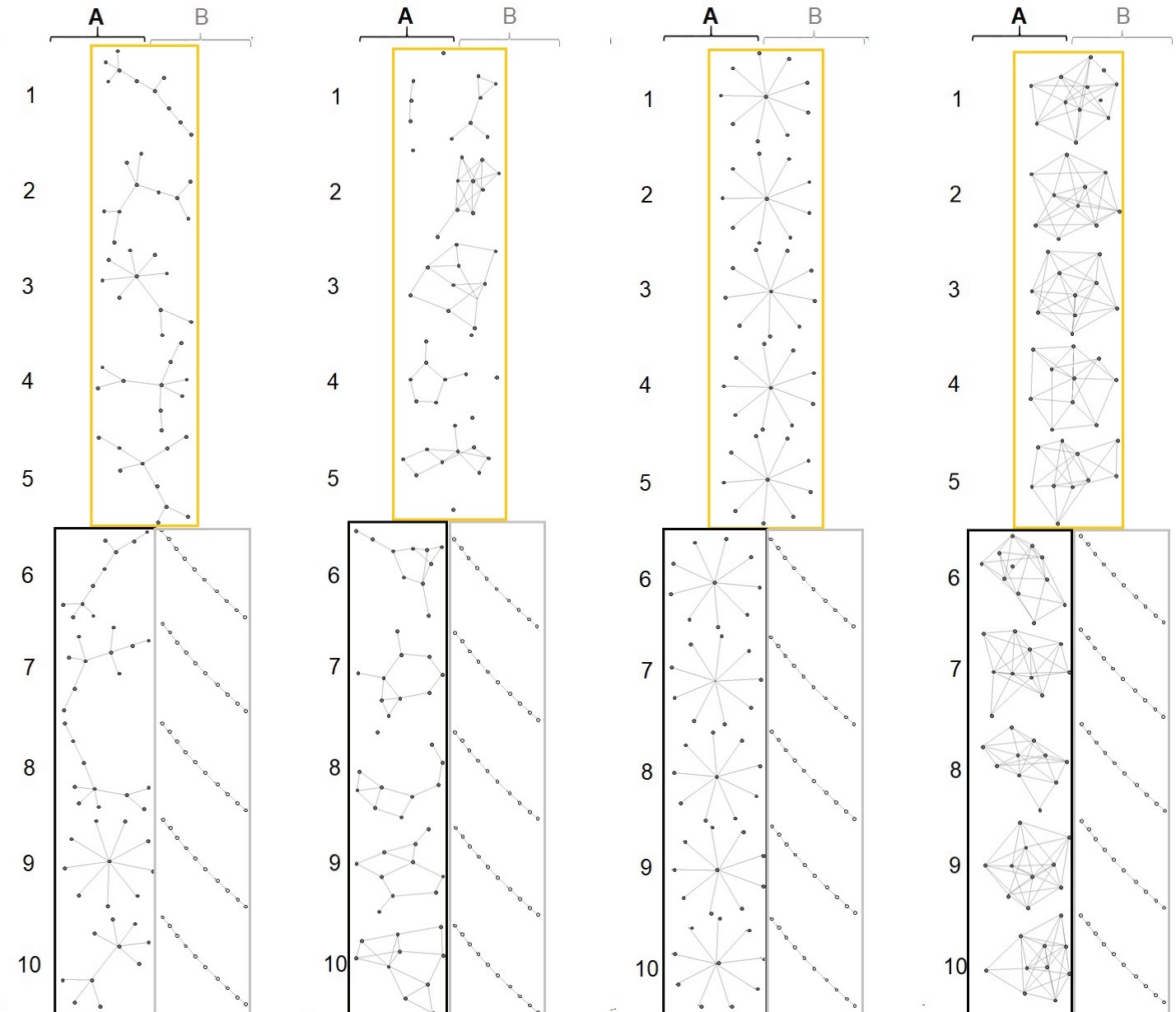

**Fig 2. Graph structures of common and phenotype A and B-specific subnetworks for scenarios 1–4, where networks in yellow, black and grey boxes are common, pheynotype A and B -specific subnetwork, respectively.**

situation 3: $\mu_A = 0, \mu_B = 1, \sigma = 3$; see the right side of Table 1). Furthermore, SAM-GS cannot perform effectively when there is no significant difference in the expression levels (i.e., situation 1: $\mu_A = 0, \mu_B = 0.3, \sigma = 1$; see the left side of Table 1). The EVD shows the poor results in overall, especially EVD cannot perform properly in the situation 3: $\mu_A = 0, \mu_B = 1, \sigma = 3$. Existing methods do not perform well in situations 1 and 3, because they are based only on the expression levels of genes. In contrast, our strategy incorporates not only the expression levels of genes but also the network structure, thereby effectively performs differential gene network identification.

**Table 1. Simulation results 1.**

|  |  | Situation 1 $\mu_A = 0, \mu_B = 0.3$ $\sigma = 1$ |  |  |  | Situation 2 $\mu_A = 0, \mu_B = 1$ $\sigma = 1$ |  |  |  | Situation 3 $\mu_A = 0, \mu_B = 1$ $\sigma = 3$ |  |  |  |
|---|---|---|---|---|---|---|---|---|---|---|---|---|---|
|  |  | SF | RD | HUB | CL | SF | RD | HUB | CL | SF | RD | HUB | CL |
| ACC | DGNdetector | **0.99** | **0.99** | **1.00** | **1.00** | **1.00** | 0.97 | **1.00** | **1.00** | 0.98 | **1.00** | **1.00** | **1.00** |
|  | SAM-GS | 0.91 | 0.93 | 0.89 | 0.88 | **1.00** | **0.99** | 0.99 | **1.00** | **0.99** | **1.00** | 0.99 | **1.00** |
|  | GSCA | 0.98 | **0.99** | 0.99 | 0.99 | 0.99 | **1.00** | 0.99 | 0.99 | 0.58 | 0.57 | 0.65 | 0.61 |
|  | EVD | 0.51 | 0.50 | 0.55 | 0.55 | 0.56 | 0.55 | 0.58 | 0.68 | 0.50 | 0.50 | 0.50 | 0.50 |
| RECALL | DGNdetector | 1.00 | 1.00 | 1.00 | 1.00 | 1.00 | 1.00 | 1.00 | 1.00 | 1.00 | 1.00 | 1.00 | 1.00 |
|  | SAM-GS | 0.83 | 0.86 | 0.78 | 0.76 | 1.00 | 1.00 | 1.00 | 1.00 | 1.00 | 1.00 | 1.00 | 1.00 |
|  | GSCA | 0.98 | 1.00 | 1.00 | 0.99 | 0.99 | 1.00 | 1.00 | 1.00 | 0.16 | 0.16 | 0.31 | 0.24 |
|  | EVD | 0.03 | 0.02 | 0.10 | 0.11 | 0.12 | 0.11 | 0.16 | 0.38 | 0.02 | 0.01 | 0.02 | 0.01 |
| PREC | DGNdetector | 0.99 | 0.99 | 0.99 | 1.00 | 0.99 | 0.95 | 0.99 | 1.00 | 0.97 | 0.99 | 0.99 | 1.00 |
|  | SAM-GS | 1.00 | 0.99 | 1.00 | 0.99 | 0.99 | 0.98 | 0.99 | 0.98 | 0.98 | 1.00 | 0.99 | 1.00 |
|  | GSCA | 0.99 | 0.98 | 0.99 | 0.99 | 0.99 | 1.00 | 0.99 | 0.99 | 0.56 | 0.59 | 0.78 | 0.74 |
|  | EVD | 0.13 | 0.06 | 0.40 | 0.46 | 0.50 | 0.53 | 0.57 | 0.91 | 0.07 | 0.06 | 0.08 | 0.033 |
| TNR | DGNdetector | 0.98 | 0.99 | 0.99 | 1.00 | 0.99 | 0.94 | 0.99 | 1.00 | 0.97 | 0.99 | 0.99 | 1.00 |
|  | SAM-GS | 1.00 | 0.99 | 1.00 | 0.99 | 0.99 | 0.98 | 0.98 | 0.99 | 0.98 | 1.00 | 0.98 | 0.99 |
|  | GSCA | 0.99 | 0.98 | 0.99 | 0.99 | 0.99 | 1.00 | 0.99 | 0.98 | 1.00 | 0.97 | 0.99 | 0.99 |
|  | EVD | 0.98 | 0.99 | 1.00 | 1.00 | 1.00 | 0.99 | 1.00 | 0.98 | 0.98 | 0.99 | 0.99 | 0.99 |
| F.measure | DGNdetector | 0.99 | 1.00 | 1.00 | 1.00 | 1.00 | 0.98 | 1.00 | 1.00 | 0.99 | 1.00 | 1.00 | 1.00 |
|  | SAM-GS | 0.90 | 0.91 | 0.86 | 0.85 | 1.00 | 0.99 | 0.99 | 1.00 | 0.99 | 1.00 | 0.99 | 1.00 |
|  | GSCA | 0.98 | 0.99 | 1.00 | 0.99 | 0.99 | 1.00 | 1.00 | 0.99 | 0.25 | 0.25 | 0.43 | 0.35 |
|  | EVD | 0.05 | 0.03 | 0.16 | 0.17 | 0.19 | 0.18 | 0.24 | 0.52 | 0.03 | 0.02 | 0.03 | 0.02 |

## Simulation study 2

Additionally, we considered the network structures of phenotype B, which were slightly different from those of phenotype A. For scenarios 1, 2, 3 and 4, the precision matrices were generated from "scale-free", "random", "hub", and "cluster" graph structures, similar to simulation study 1. For the five precision matrices of the phenotype B-specific subnetwork, we replaced the randomly selected nonzero entries of the precision matrices of phenotype A with zero entries. In other words, we define the precision matrices of phenotype B ($\Omega_B^{nw}$) by replacing 50% of the nonzero entries of $\Omega_A^{nw}$ with zero for $nw = 1, \ldots, 5$. We then generated the expression levels of phenotypes A and B, similar to those in simulation study 1. The evaluation results are listed in Table 2. Our technique exhibited outstanding performance in differentially regulated gene network identification, whereas SAM-GS and GSCA could not perform well in situations where there is no considerable difference in expression levels or a large variance, respectively. Based on these results, we expect our method to be a useful tool for differential gene network analysis.

## COVID-19 severity specific gene network identification

To reveal the COVID-19 severity-specific molecular interplay in the Japanese population, we performed differential gene network analysis based on whole blood RNA-seq data from the Japan COVID-19 Task Force [18], where the dataset comprises 5,985 genes and 473 samples. The 473 samples were annotated with four levels of phenotype severity, i.e., "Most severe (patients in intensive care unit or requiring intubation and ventilation)", "Severe (others requiring oxygen support)", "Mild (other symptomatic patients)", and "Asymptomatic (without COVID19 related symptoms)" [18]. The 368 samples with levels "Most severe" and "Severe" are defined as COVID-19 severe samples, and 105 samples composed of "Mild", and "Asymptomatic" levels are defined as non-severe samples.

**Table 2. Simulation results 2.**

| | | Situation 1 $\mu_A = 0, \mu_B = 0.3$ $\sigma = 1$ | | | | Situation 2 $\mu_A = 0, \mu_B = 1$ $\sigma = 1$ | | | | Situation 3 $\mu_A = 0, \mu_B = 1$ $\sigma = 3$ | | | |
|---|---|---|---|---|---|---|---|---|---|---|---|---|---|
| | | SF | RD | HUB | CL | SF | RD | HUB | CL | SF | RD | HUB | CL |
| ACC | DGNdetector | 0.94 | 0.94 | 0.93 | 0.98 | 0.99 | 1.00 | 0.99 | 0.98 | 1.00 | 0.99 | 0.99 | 0.99 |
| | SAM-GS | 0.88 | 0.90 | 0.85 | 0.88 | 0.99 | 1.00 | 0.99 | 0.99 | 0.98 | 1.00 | 0.99 | 0.99 |
| | GSCA | 0.84 | 0.84 | 0.89 | 0.96 | 0.87 | 0.90 | 0.90 | 0.98 | 0.53 | 0.52 | 0.53 | 0.54 |
| | EVD | 0.71 | 0.70 | 0.73 | 0.70 | 0.81 | 0.86 | 0.82 | 0.80 | 0.50 | 0.51 | 0.50 | 0.51 |
| RECALL | DGNdetector | 0.89 | 0.90 | 0.87 | 0.97 | 1.00 | 1.00 | 1.00 | 1.00 | 1.00 | 1.00 | 1.00 | 1.00 |
| | SAM-GS | 0.77 | 0.79 | 0.72 | 0.77 | 1.00 | 1.00 | 1.00 | 1.00 | 1.00 | 1.00 | 1.00 | 1.00 |
| | GSCA | 0.69 | 0.69 | 0.78 | 0.92 | 0.75 | 0.82 | 0.82 | 0.97 | 0.08 | 0.06 | 0.06 | 0.09 |
| | EVD | 0.44 | 0.42 | 0.47 | 0.42 | 0.62 | 0.74 | 0.66 | 0.60 | 0.02 | 0.03 | 0.02 | 0.05 |
| PREC | DGNdetector | 0.99 | 0.98 | 0.99 | 0.99 | 0.98 | 0.99 | 0.98 | 0.97 | 0.99 | 0.99 | 0.99 | 0.99 |
| | SAM-GS | 0.99 | 1.00 | 0.98 | 1.00 | 0.99 | 0.99 | 0.98 | 0.99 | 0.97 | 1.00 | 0.98 | 1.00 |
| | GSCA | 0.99 | 0.99 | 0.99 | 1.00 | 0.98 | 0.98 | 0.97 | 0.99 | 0.30 | 0.25 | 0.26 | 0.39 |
| | EVD | 0.92 | 0.94 | 0.95 | 0.86 | 0.98 | 0.99 | 0.99 | 0.99 | 0.10 | 0.14 | 0.08 | 0.18 |
| TNR | DGNdetector | 0.99 | 0.98 | 0.99 | 0.98 | 0.97 | 0.99 | 0.98 | 0.97 | 0.99 | 0.99 | 0.98 | 0.99 |
| | SAM-GS | 0.99 | 1.00 | 0.98 | 0.99 | 0.99 | 0.99 | 0.98 | 0.99 | 0.97 | 1.00 | 0.97 | 0.98 |
| | GSCA | 0.99 | 0.99 | 0.99 | 1.00 | 0.99 | 0.98 | 0.97 | 0.98 | 0.98 | 0.98 | 0.99 | 0.99 |
| | EVD | 0.98 | 0.98 | 0.99 | 0.99 | 0.97 | 0.99 | 0.99 | 0.99 | 0.98 | 0.99 | 0.99 | 0.98 |
| F.measure | DGNdetector | 0.93 | 0.94 | 0.92 | 0.98 | 0.99 | 1.00 | 0.99 | 0.99 | 1.00 | 1.00 | 0.99 | 1.00 |
| | SAM-GS | 0.85 | 0.88 | 0.81 | 0.84 | 1.00 | 1.00 | 0.99 | 1.00 | 0.99 | 1.00 | 0.99 | 0.99 |
| | GSCA | 0.80 | 0.80 | 0.86 | 0.95 | 0.84 | 0.89 | 0.88 | 0.98 | 0.12 | 0.10 | 0.10 | 0.15 |
| | EVD | 0.57 | 0.56 | 0.61 | 0.54 | 0.73 | 0.84 | 0.77 | 0.73 | 0.04 | 0.05 | 0.03 | 0.07 |

We focused on the molecular interplay of the identified genes in the study conducted by the Japan COVID-19 Task Force [18]. We estimated severe- and non-severe-specific gene networks based on their expression levels in the severe and non-severe samples, respectively. We considered a linear regression model to describe the gene network, where the response and predictor variables were the expression levels of the target and regulator genes, respectively. That is, we estimated 5,985 models for 5,985 target genes using the lasso [29]. We focused on severely specific gene networks. The gene network consisted of 5,985 genes, 675,337 edges, and two subnetworks. We considered the edges with the top 0.1% largest absolute edge weights (i.e., 676 edges) and their networks, where 667 genes constructed 200 subnetworks with sizes of 2∼46 genes. Differential gene network analysis by DGNdetector was performed for 58 subnetworks with more than two edges. Our method revealed 46 differentially regulated gene networks between severe and non-severe samples, based on 500 permutations and $p.value < 0.01$.

We then focused on the genes identified by the Japan COVID-19 Task Force (Supplementary Data 2–10 of [18]), named J-COVID19 markers, and investigated their molecular interplay by our computational network biology method. From the identified 46 differentially regulated gene networks, we extracted 15 subnetworks that consisted of at least one of the J-COVID19 markers. Fig 3 shows the differentially regulated networks of the COVID-19 markers between the severe and non-severe samples.

As shown in Fig 3, severe samples showed relatively dense gene networks compared with non-severe samples. We focused on three large-scale networks, marked 1, 2, and 3.

- **Subnetwork 1**

    Subnetwork 1 consists of 25 genes, where only CIITA is the COVID-19 marker. The interplay with HLA class II dominated the identified differentially regulated gene network, i.e., Subnetwork 1, in the severe samples. Their interplay became weaker and/or disappeared

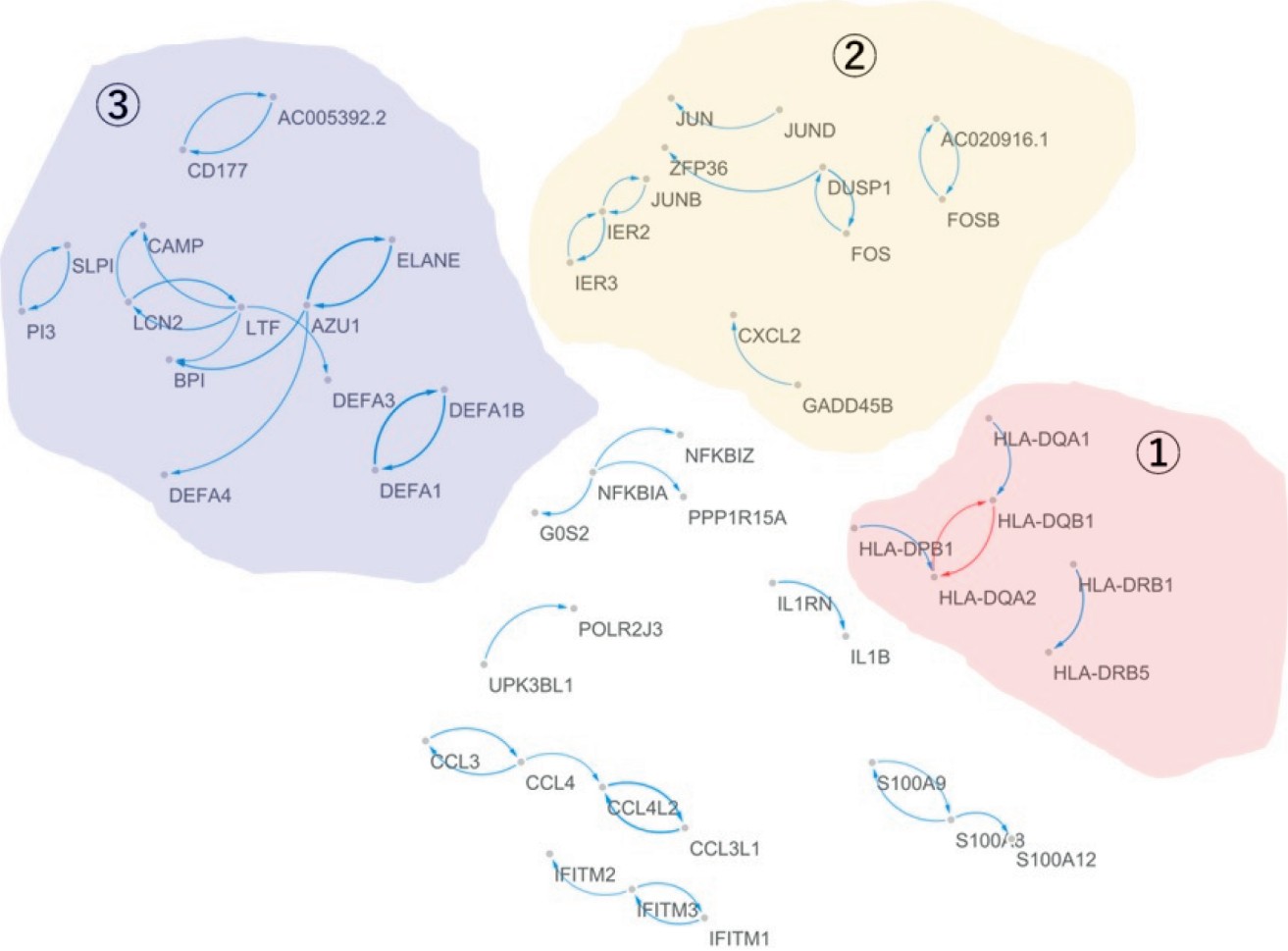

**Fig 3. Differentially regulated gene network between severe and non-severe samples.** Edge thickness represents the strength of edge, color indicates sign of the effect (red: "-" and blue: "+"), and arrow (X → Y) indicates that gene X regulates gene Y.

in the gene network of non-severe samples. Furthermore, the molecular interplay between HLA class II and the COVID-19 marker CIITA was observed in only severe samples. The interplay between HLA class II and CD74 was also considered a severity characteristic. Thus, it can be considered that the identified Subnetwork 1 is a severity-specific molecular marker and provides crucial clues to uncover the mechanism of COVID-19 severity.

- **Subnetwork 2**

  Subnetwork 2 consists of 15 genes, among which three genes (i.e., CXCL1, CXCL2, GADD45B) are the J-COVID19 markers. The interaction between the CXCL family (CXCL1, CXCL2, and CXCL8) is also considered a COVID-19 severity-specific molecular interplay. This interplay disappeared in the network of non-severe samples.

- **Subnetwork 3**

  Subnetwork 3 comprised 26 genes of which 21 genes were the J-COVID19 markers identified by the Japan COVID-19 Task Force and only five genes (AC005392.2, ST3GAL4,

ADGRE3, C15orf48, PI3) were newly suggested as regulator and target genes of the markers. This finding implies that Subnetwork 3 can be considered as a gene network of COVID-19 markers.

We focus on Subnetwork 1, where HLA class II is the main player. Human leukocyte antigen (HLA) molecules play key roles in the adaptive immune system by sending signals regarding the health status of cells to the immune system [30].

The molecular interplay between the HLA class II and CIITA in COVID-19 has been demonstrated as follows [20]: CIITA, a master transcriptional regulator, facilitates the peptide-loading machinery and cell surface expression of HLA class II complexes, and has been used to interrogate the HLA class II immunopeptidome of SARS-CoV-2 infected cells and tumors [20]. Weingarten-Gabbay et al. [20] suggested the use of CIITA over expression to infer the HLA-II immunopeptidome in cancer cells and viruses. Bruchez et al. [19] showed that the antiviral activities of CIITA and CD74 protect against coronaviruses. These results imply that the identified differentially regulated gene network, i.e., subnetwork 1 consisting of HLA class II, CIITA, and CD74 may be a key marker for uncovering mechanism underlying COVID-19 severity. Furthermore, the association of HLA class II with the susceptibility, severity, and progression of COVID-19 has been demonstrated in various studies. The crucial role of HLA molecules in the immune response and the molecular variability of HLA alleles related to the different rates of infection and patients following COVID-19 have been demonstrated [21]. The use of HLA testing in clinical trials and the combination of HLA typing with COVID-19 testing has also been suggested to more rapidly identify predictors of viral severity [21]. The influence of HLA genotype on the severity of COVID–19 infection in European populations was investigated in a previous study; a significant difference in the allele frequency of HLA–DRB1*04:01 in the severe patient group [22]. Although many studies have focused on the association between HLA class II and COVID-19 [20–22], our results suggest that the interplay between HLA class II and its regulator and/or target genes (e.g., CIITA and CD74) may play a crucial role in COVID-19 severity.

To identify the biological pathways and functions of Subnetwork 1 (i.e., the interplay between HLA class II, CIITA and CD74), we performed a Kyoto Encyclopedia of Genes and Genomes (KEGG) pathway analysis (https://www.genome.jp/kegg/). Fig 4 shows the KEGG enrichment analysis results.

The KEGG pathway analysis revealed that "hsa04612:Antigen processing and presentation", "hsa05330:Allograft rejection", "hsa05332:Graft-versus-host disease", "hsa04940:Type I diabetes mellitus" and "hsa05320:Autoimmune thyroid disease" are the top 5 enriched pathways for the genes in the Subnetwork 1. Chen et al. [31] showed that the discriminative genes in immune cells (i.e., B cell) in healthy control, severe, and critical COVID-19 patients are enriched in the "hsa04612:Antigen processing and presentation" pathway. Furthermore, "hsa04612:Antigen processing and presentation" was identified as the enriched pathway of the genes that show significant differential changes in hubness (number of connections) between different stages of COVID-19 (i.e., healthy, moderate, severe, convalescence stage) [32]. The pathway "hsa05330:Allograft rejection" was identified for the genes downregulated in patients with severe infection compared with those with mild infection and SARS-CoV-2-negative control individuals [33]. Szyda et al. [34] revealed that "hsa05330:Allograft rejection" is associated with resistance to COVID-19 infection and the pathway comprises several immune system components for self-versus-non-self recognition. A recent study [35] demonstrated an association between type 1 diabetes mellitus and increased morbidity and mortality rates during COVID-19 infection, suggesting that vaccination for these patients should be prioritized. Furthermore, several studies have demonstrated an association between diabetes mellitus, disease severity, and prognosis in patients with COVID-19 [36,37].

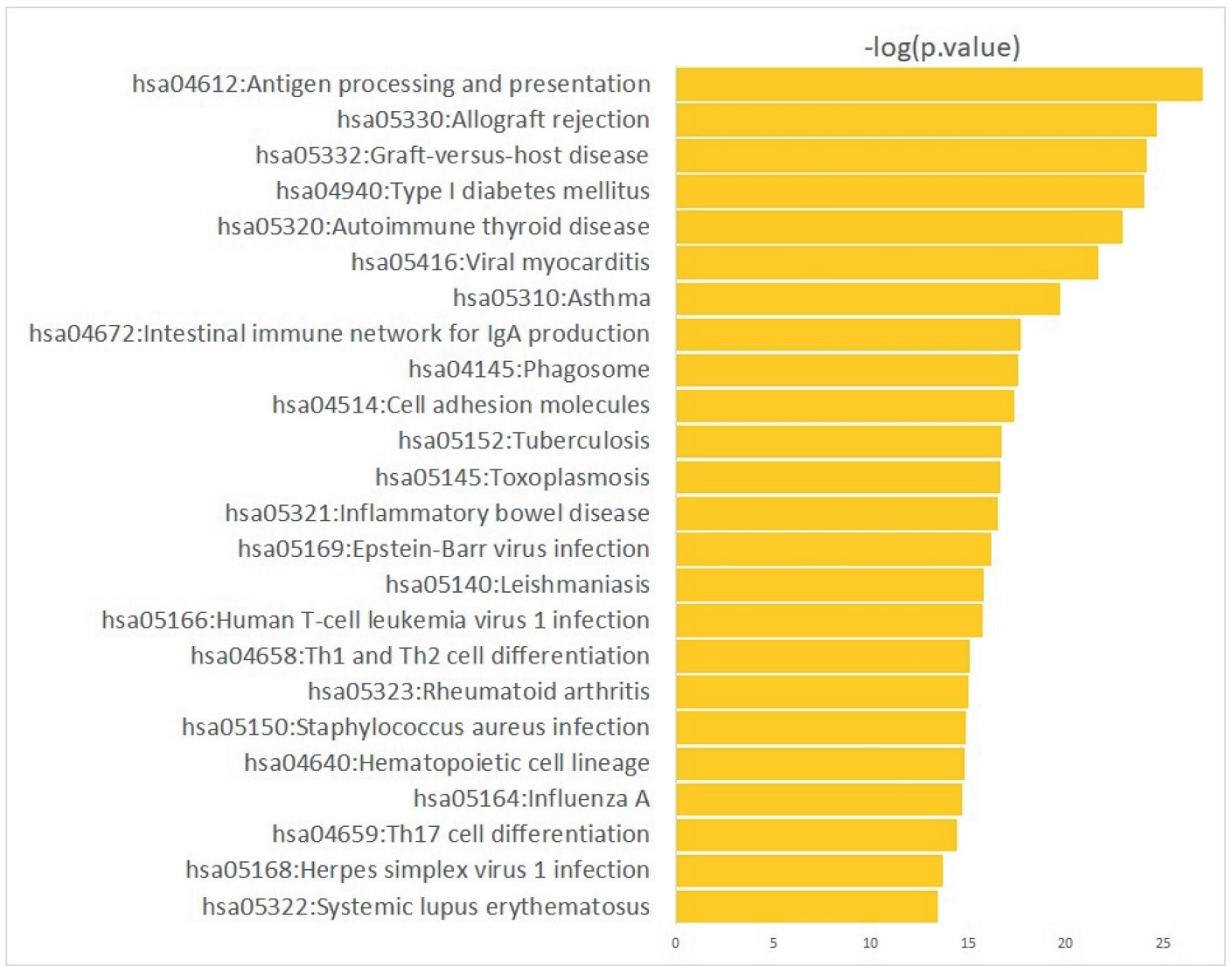

**Fig 4. KEGG pathway analysis of the genes in the Subnetwork 1, which is differentially regulated gene network in severe samples.**

This implies that HLA class II and CIITA are crucial markers for understanding COVID-19 severity. Furthermore, the interplay between HLA class II, CIITA, and CD74 may play a key role in the severity of COVID-19. Based on our results and the literature, we suggest that controlling the molecular interplay in Subnetwork 1 (i.e., the interplay between HLA class II, CIITA, and CD74) and the enriched pathways provides crucial clinical insights into the mechanism of COVID-19 severity.

## Conclusions

We aimed to elucidate the mechanism underlying COVID-19 severity based on gene network analysis. We developed a computational network biology strategy to identify differentially regulated gene networks between severe and non-severe COVID-19 samples. In our strategy, we describe the gene networks using the probability density function based on mean vectors

of expression levels and network structures (i.e., estimated precision matrices). We then measured the dissimilarity of gene networks based on the Kullback–Leibler divergence. The developed method incorporates comprehensive information about gene networks, which refers to not only the expression levels of genes, but also the network structure, and thus our strategy can provide informative results for identifying differentially regulated gene networks.

To illustrate the efficiency of the proposed strategy, we conducted Monte Carlo simulations and demonstrated its outstanding performance. We applied our strategy to whole-blood RNA-seq data from the Japan COVID-19 Task Force and identified a differentially regulated gene network between COVID-19 severe and non-severe samples, focusing on COVID-19 severe-specific molecular interplay. The proposed computational network biology strategy revealed the molecular interplay between HLA class II and the identified COVID-19 markers CIITA/CD74 as a COVID-19 severity sepcific marker in the Japanese population. These results are strongly supported by those of the previous studies. Our results suggest that not only HLA class II but also its molecular interplay with CIITA/CD74 may be a key marker for uncovering the mechanism underlying COVID-19 severity. Suppression and activation of this molecular interplay may provide crucial clues to address COVID-19 severity.

The novelty of our study is trying to uncover COVID-19 severe specific biomarkers based on the molecular interplays, while previous studies focused on the abnormalities of each single gene. In order to achieve this, we proposed the novel computational strategy for identifying differentially regulated gene networks between COVID-19 severe and non-severe samples based on comprehensive information of gene networks (i.e., not only expression levels of genes but also network structure).

Although our method provided effective and biologically reliable results for differentially regulated gene network identification, the proposed DGNdetector suffers from the computation complexity, because our strategy is based on permutation framework. In the future work, we will extend our strategy to time effective method based on parametric approach instead of permutation frame work.

In Section of COVID-19 severity specific gene network identification, we perform gene networks analysis for 368 sever and 105 non-severe samples. The analysis of the different sample sizes of phenotypes can be considered as one of limitations of our studies, because the different sample sizes may raise biases on the reliability. To avoid the bias from the different sample sizes, bootstrap strategy or randomly selected samples from 368 sever samples -based analysis can be considered as another future works of our studies.

## Acknowledgments

This research used the computational resources of Super Computer System, Human Genome Center, Institute of Medical Science, University of Tokyo.

## Author contributions

**Conceptualization:** Heewon Park, Satoru Miyano.

**Formal analysis:** Heewon Park.

**Methodology:** Heewon Park.

**Supervision:** Satoru Miyano.

**Writing – original draft:** Heewon Park.

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
