## [Decision Letter · Decision Letter 0]

1 Dec 2024

PONE-D-24-41990Computational network biology analysis revealed COVID-19 severity markers: Molecular interplay between HLA-2 with CIITAPLOS ONE

Dear Dr. Park,

Thank you for submitting your manuscript to PLOS ONE. After careful consideration, we feel that it has merit but does not fully meet PLOS ONE’s publication criteria as it currently stands. Therefore, we invite you to submit a revised version of the manuscript that addresses the points raised during the review process.

Efforts to set up new experimental approaches to gain new evidence that could enhance our understanding of the mechanisms underpinning the severe outcome of COVID-19 are valuable. The authors aimed to tackle the challenge through computational analysis of biological networks, developing a new strategy DGN detector on severe and non-severe samples of COVID-19. Regarding this point, there is a wide discrepancy in the sample size. Could the authors explain this choice and if it might raise biases on the reliability of their statistical analyses?

The authors should add the bibliographical references missing in several phrases throughout the text. At the end of the introduction, the authors should summarize the results and briefly outline their meanings. Please avoid indicating the structure of the manuscript. The writing includes excessive technical jargon, making it difficult for readers to follow the strategy employed and understand the implications of the findings.

The conclusion appears vague. Could the authors delve into the novelty of their findings?

We look forward to receiving your revised manuscript.

Kind regards,

Elisabetta Pilotti

Academic Editor

PLOS ONE

Journal Requirements:

2. Thank you for stating the following financial disclosure: [The work of Heewon Park was supported by NRF (RS-2023-00276559).

This research was also supported by AMED under Grant Number 23tk0124003h0001, 24tk0124003h0002, JSPS KAKENHI Grant Number 24H00009/22H03692 and JST Grant Number JPMJCR20H2].

Reviewers' comments:

Reviewer's Responses to Questions

**Comments to the Author**

1. Is the manuscript technically sound, and do the data support the conclusions?

Reviewer #1: Yes

Reviewer #2: Yes

2. Has the statistical analysis been performed appropriately and rigorously? 

Reviewer #1: No

Reviewer #2: Yes

3. Have the authors made all data underlying the findings in their manuscript fully available?

Reviewer #1: Yes

Reviewer #2: Yes

4. Is the manuscript presented in an intelligible fashion and written in standard English?

Reviewer #1: Yes

Reviewer #2: Yes

5. Review Comments to the Author

Reviewer #1: The topic of the article is very interesting. However, the article requires some revision. Some of the comments are presented as:

1) Avoid writing abbreviations in the abstract

2) The abstract should be comprehensive that indicate the clear picture of the work

3) The introduction needs more details based on the coronavirus literature

4) What is the novelty of this work

5) Highlight the novelty in the introduction section

6) Provide the organization of the paper at the end of introduction section

7) What are the merits and demerits of the proposed scheme?

8) Compare the results with state of art method

9) What is the stopping criteria of the scheme?

10) Update the conclusions

11) The introduction can be updated by referring to more recent and relevant articles such as

a. "Bio inspired heuristic computing scheme for the human liver nonlinear model." Heliyon 10.7 (2024).

b. "A computational framework to solve the nonlinear dengue fever SIR system." Computer Methods in Biomechanics and Biomedical Engineering 25.16 (2022): 1821-1834.

c. "An artificial neural network approach for the language learning model." Scientific reports 13.1 (2023): 22693.

Reviewer #2: The research foresees a significant contribution considering that the topic is topical, the methods are correct and the results are within the expected range, taking into account the review of the background described in the research.

6. PLOS authors have the option to publish the peer review history of their article (what does this mean?). If published, this will include your full peer review and any attached files.

Reviewer #1: No

Reviewer #2: No

---

## [Author Response · Author response to Decision Letter 1]

15 Dec 2024

We would like to thank you for insightful comments and suggestions.

We carefully incorporated the comments and suggestions in the revised version of our manuscript.

Our point by point responses will be attached.

---

## [Editor Report · Decision Letter 1]

18 Dec 2024

PONE-D-24-41990R1Computational network biology analysis revealed COVID-19 severity markers: Molecular interplay between HLA-2 with CIITAPLOS ONE

Dear Dr. Park,

Thank you for submitting your manuscript to PLOS ONE. After careful consideration, we feel that it has merit but does not fully meet PLOS ONE’s publication criteria as it currently stands. Therefore, we invite you to submit a revised version of the manuscript that addresses the points raised during the review process.

In the abstract, I recommend using the acronym COVID-19 instead of repeatedly writing "coronavirus disease 2019." The same applies to the term “Human Leukocyte Antigen.” They only need to be explained at the first mention in the introduction.

As in the first letter, I again suggested the authors end the introduction with the novelties of their experimental strategies and the importance of the findings and eliminate the paragraph from lines 112-117. There is no need to specify the structure of the manuscript.

On line 83, I propose rephrasing “It can be through our results that…” to improve clarity.

Moreover, I encourage the authors to succinctly convey the novelties of their studies in one concise sentence, avoiding the breakdown into multiple points. I also suggest avoiding statements such as “can effectively reveal the COVID-19 severity mechanisms of COVID-19” (line 104) since the authors rightly underlined the limitations of their study.

Could the author delve into the description of samples, clarifying what constitutes the severe and non-severe groups?

At the end of the conclusions, the paragraph from lines 400 to 408 is somewhat convoluted and feels redundant in expressing key claims. A more straightforward presentation of these ideas would enhance readability and comprehension.

Lastly, I noticed some errors in typing, so please amend them.

We look forward to receiving your revised manuscript.

Kind regards,

Elisabetta Pilotti

Academic Editor

PLOS ONE
---

## [Author Response · Author response to Decision Letter 2]

26 Jan 2025

We would like to thank you for insightful comments and suggestions.

We carefully incorporated the comments and suggestions in the revised version of our manuscript.

Our point by point responses will be attached.

---

## [Editor Report · Decision Letter 2]

29 Jan 2025

Computational network biology analysis revealed COVID-19 severity markers: Molecular interplay between HLA-2 with CIITA

PONE-D-24-41990R2

Dear Dr. Park,

We’re pleased to inform you that your manuscript has been judged scientifically suitable for publication and will be formally accepted for publication once it meets all outstanding technical requirements.

Kind regards,

Elisabetta Pilotti

Academic Editor

PLOS ONE
---

## [Editor Report · Acceptance letter]

PONE-D-24-41990R2

PLOS ONE

Dear Dr. Park,

I'm pleased to inform you that your manuscript has been deemed suitable for publication in PLOS ONE. Congratulations! Your manuscript is now being handed over to our production team.

Kind regards,

on behalf of

Dr. Elisabetta Pilotti

Academic Editor

PLOS ONE